# The impact of a competitive event and the efficacy of a lactic acid bacteria-fermented soymilk extract on the gut microbiota and urinary metabolites of endurance athletes: An open-label pilot study

Mina Fukuchi[1], Masaaki Sugita[2,¤a], Makoto Banjo[2,¤b], Keisuke Yonekura[3], Yasuhiro Sasuga[1]*

1 Hachioji Center for Research and Development, B&S Corporation Co., Ltd., Tokyo, Japan, 2 Faculty of Education, Mie University, Mie, Japan, 3 B&S Corporation Co., Ltd., Tokyo, Japan

¤a Current address: Faculty of Sport Science, Nippon Sport Science University, Tokyo, Japan
¤b Current address: Asahigaoka Elementary School, Gifu, Japan
* y-sasuga@bandscorp.jp

**Data Availability Statement:** Fastq files are deposited in the DDBJ database under the accession number DRA011638 with h BioProject

## Abstract

Diet and exercise can alter the gut microbiota, but recent studies have assessed the impact of athletic competition on gut microbiota and host metabolites. We designed an open-label pilot study to investigate the effects of both official competition and a multi-strain lactic acid bacteria-fermented soymilk extract (LEX) on the gut microbiota in Japanese college endurance athletes. The analysis of fecal 16S rRNA metagenome and urinary metabolites was used to identify changes in gut microbiota composition and host metabolism. When the fecal microbiota were investigated before and after a race without using of a supplement (pre-observation period), there was an increase in the phylum *Firmicutes* and decrease in *Bacteroidetes*. However, no changes in these phyla were seen before and after a race in those who consumed LEX. Before and after LEX ingestion, changes in urinary metabolites included a significant reduction in yeast and fungal markers, neurotransmitters, and mitochondrial metabolites including the TCA cycle. There were several correlations between urinary metabolites and the composition of fecal microbiota. For example, the level of tricarballylic acid was positively correlated with the composition ratio of phylum *Firmicutes* (Pearson's $r = 0.66$; $p < 0.01$). The bacterial species *Parabacteroides distasonis* was also found to correlate moderately with several urinary metabolites. These findings suggest two possibilities. First, endurance athletes experience significant fluctuations in gut microbiota after a single competition. Second, LEX ingestion may improve yeast and fungal overgrowth in the gastrointestinal tract and enhancing mitochondrial metabolic function.

ID PRJDB11304 and BioSample IDs SAMD00283406-SAMD00283454.

**Funding:** Funding: This work was supported by B&S Corporation Co. Ltd. The funder provided support in the form of salaries for authors [MF, YK and YS], but did not have any additional role in the study design, data collection and analysis, decision to publish, or preparation of the manuscript. The specific roles of these authors are articulated in the 'author contributions' section.

**Competing interests:** Competing interests: The authors have read the journal's policy and the authors of this manuscript have the following competing conflicts. The test article used for this study was manufactured and marketed by B&S Corporation Co. Ltd. Authors [MF, YK and YS] are paid employees of B&S Corporation Co. Ltd. There are no patents or products in development to declare. This does not alter our adherence to PLOS ONE policies on sharing data and materials.

## Introduction

The gut microbiota is essential for health, playing a role in nutrient uptake, vitamin synthesis, energy harvest, inflammatory modulation, and host immune response [1, 2]. The use of next-generation sequencing techniques has greatly expanded our knowledge of microbiota composition and its relationship to diseases [3]. Numerous intrinsic and extrinsic factors can affect the gut microbiota, resulting in a highly dynamic and complex gut environment. The diet is the main modifying factor affecting the composition of human microbiota, and dietary components act as substrates for microbial metabolism, influencing both microbiome composition and function [4]. Additionally, recent studies suggest the ability of physical exercise to induce changes in the gut microbiota, which may also affect exercise performance. For instance, when compared with sedentary control groups, athletes had relative increases in pathways (e.g., amino acid, antibiotic biosynthesis, carbohydrate metabolism) and fecal metabolites (e.g., short-chain fatty acids) associated with enhanced muscle turnover [5].

Several studies showed that elite athletes and those who train frequently have higher bacterial species richness (α-diversity) than those with a sedentary lifestyle or low fitness level [6–8]. An increase in the symbiotic species *Akkermansia muciniphila* and *Faecalibacterium prausnitzii* was observed in athletes and highly active individuals [6, 9].

Excessive exercise stresses the gastrointestinal (GI) tract and increases the likelihood of multiple symptoms associated with the disruption of the gut microbiota and decreased performance [10]. Supplements aimed at improving the intestinal environment, including probiotics, have traditionally focused on the health status of athletes in terms of reducing exercise-induced stress, improving the host immunity, reducing the symptoms of GI and upper respiratory tract infections [11]. In addition to probiotic products, various other supplements such as fermented bacterial products are expected to improve GI tract symptoms. The extract of multi-strain lactic acid bacteria (LAB)-fermented soy milk (LEX) is one of them, and the effect of improving the metabolites derived from gut microbiota has been reported [12]. Another study demonstrated that LEX ingestion can prevent colon cancer and activate intestinal immunity [13, 14].

Understanding whether gut microbiota and its environment play a vital role in athletic performance and daily conditioning is particularly interesting to athletes. Additionally, such knowledge can bring benefits to human health. Further studies are needed to understand the daily changes in gut microbiota and host metabolites or beneficial food effects on athletes' day-to-day health.

Endurance exercises are activities, which are performed during longer time intervals and used aerobic metabolism. They can be defined as long-term cardiovascular exercise and include activities such as running, cross-country skiing, cycling, aerobic exercise, or swimming. Physiological adaptations to endurance exercise include correcting electrolyte imbalances, increases in systemic inflammatory responses, and decreases in glycogen storage, oxidative stress, intestinal permeability, and muscle damage [15]. Endurance athletes are more likely to have the upper respiratory tract infections and GI disorders, including increased permeability of the gastrointestinal epithelial wall, known as "leaky gut," disruption of mucus thickness, and increased bacterial migration [16].

We conducted a pilot study modeled on long-distance runners to investigate changes in gut microbiota during excessive exercise and competition. Furthermore, we investigated the effects of LEX ingestion on the gut microbiota and urinary metabolites as an example of a supplement for affecting GI environment. Here, we use fecal microbe metagenome sequencing and urinary biological metabolome analysis to characterize changes before and after race and LEX ingestion on long-distance runners.

## Materials and methods

### Study design and participants

This was an open-label study to evaluate the impact of both official sports competition and the efficacy of LEX on endurance athletes. This study enrolled Japanese college long-distance runners aged 19 to 21 years old and a continuous period was selected so that there was one official race for each observation period. A total of 13 participants, nine males and four females, were recruited. Those who were allergic to soybeans, a raw material of the test article, were excluded. The study consisted of two 4-week periods, where each four weeks, the participants competed in one race. The first four-week pre-observation period (before LEX ingestion) measured changes in the gut microbiota due to athletic performance alone. The second four-week period examined the effect of daily LEX ingestion on gut microbiota.

Furthermore, to investigate the effect of LEX ingestion on host metabolites, we observed changes in urinary metabolites before and after the LEX ingestion. The study was conducted according to the Declaration of Helsinki. Informed consent was obtained from each participant and ethical approval was obtained from the Ethics Committee of Faculty of Education, Mie University (registration number: 2016–4, Mie, Japan).

### Test supplement

The test supplement was a commercially available LAB dietary supplement (brand name LAC-TIS) that was derived from a multi-strain LAB-fermented soymilk (LEX) extract consisting of sixteen LAB strains (*Lacticaseibacillus paracasei* R0101, R0301, R0401, R0601, R0701, R0901, R1001, R1402, R1502, R1602, *Lactiplantibacillus plantarum* R0502, R0801, R1101, Y1201, *Levilactobacillus brevis* R0201, R1305). LAB was cultivated in soymilk on the order of $10^{12}$ bacteria/g, and then extracted with ethanol, and clear extract was added to the product, as equivalent ca. $10^{10}$ bacteria/mL with lactic acid and citric acid. The test article was a 10 mL volume liquid, and participants ingest it twice in the morning and evening before meals during the LEX ingestion period. The test article was obtained from B&S Corporation Co., Ltd. (Tokyo, Japan).

### Procedure

Participants were recruited from university track and field clubs, and 13 long-distance runners were enrolled from October 20 to December 16, 2016. The participants were provided with the test article, four feces collection kits, and two urine collection kits and were instructed to ingest the test supplement twice (10 mL/once) daily for four weeks after a four-week pre-observation period.

An official race was held twice during the entire observation period, and the schedule was set so that a race would occur in the middle of both the pre-observation period and the LEX-ingestion period. The official races were the second Long-distance challenge race (2016. 11. 12, Aichi, Japan, Organizer: Aichi Association of Athletics) and the 78th The Inter-University Athletic Unions of Tokai Ekiden Road Relay Championship (2016. 12. 4, Aichi, Japan, Organizer: The Inter-University Athletic Union of Tokai), which were held during the pre-observation period and the LEX ingestion period, respectively. The primary outcome was to measure the composition of fecal microbiota and urinary metabolites. Fig 1A shows the timing of races and sample collection during the test period. The composition of fecal microbiota was measured four times: before a race and without supplementation in a pre-observation period (PRE_b), at the end of the pre-observation period (post-race) with no supplementation (PRE_a), before a race in the LEX-ingestion period (POST_b), and at the end of the LEX-

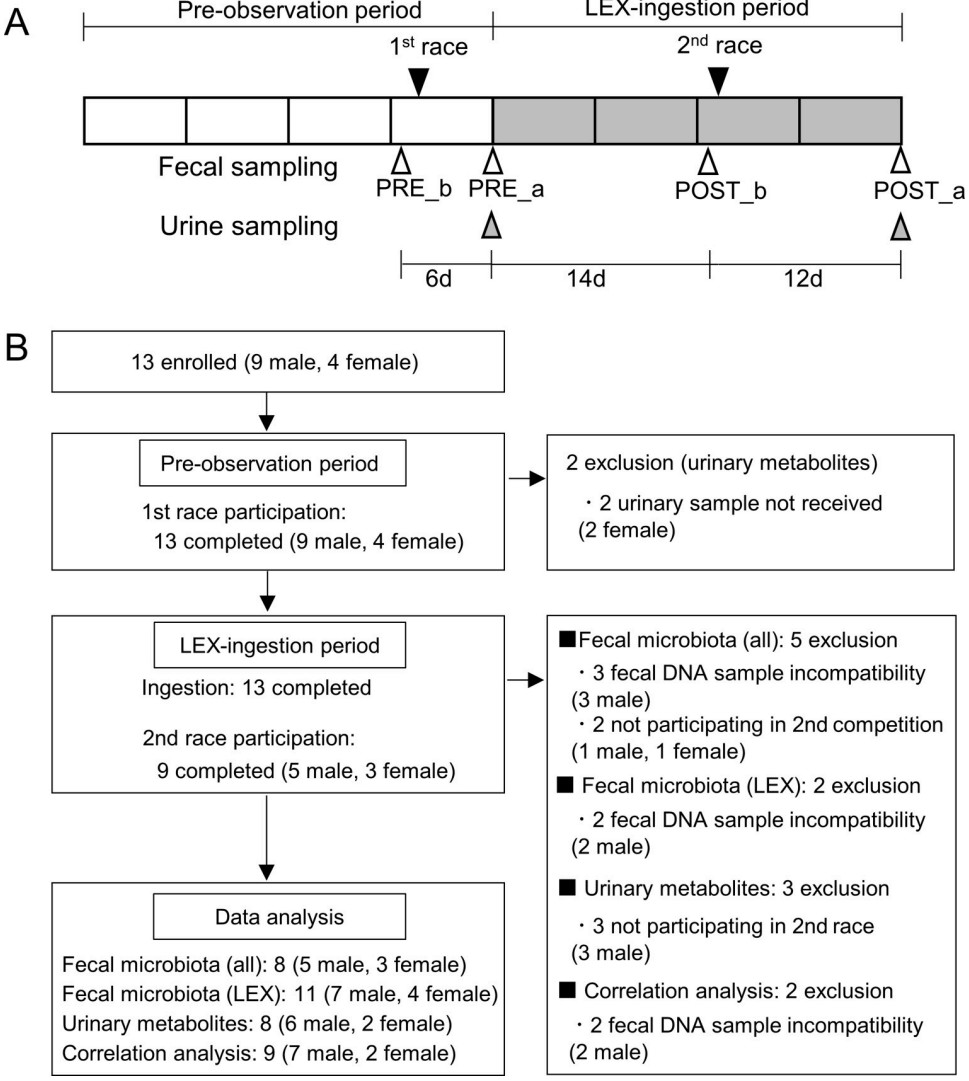

**Fig 1. Study overview.** (A) Race schedule and sampling timing of feces and urine. (B) Participant-tracking flowchart.

ingestion (post-race) period (POST_a). Additionally, urinary metabolites were measured twice at PRE_a and POST_a to evaluate the efficacy of LEX ingestion. During the observation period, participants wrote a diary about their physical condition during practice, total training time, mileage, condition of breakdown, and exercise intensity to evaluate exercise specificity. The exercise intensity was entered in 11 levels from 0 to 10 (0: practice break, 1: low, 10: high), and the same evaluation was given to the exercise load of the competition when participating in the races.

## Bacterial DNA isolation and 16S rRNA sequencing

Fecal samples were self-collected in polyethylene sample collection containers, placed in a freezer immediately, transport to lab while frozen and stored at -80°C for further analysis. Total genome DNA from fecal samples was extracted using a commercial DNA extraction kit (ISOFECAL for Beads Beating, NIPPON GENE CO., LTD., Tokyo, Japan) according to the manufacturer's instructions and stored at −30°C. The V3–V4 region of the bacterial 16S rRNA

gene was amplified by PCR using universal primers (forward: 5′– ACACTCTTTCCCTACAC–GACGCTCTTCCGATCTNNNNNCCTACGGGNGGCWGCAG–3′; reverse: 5′–GTGACTGGAGTT–CAGACGTGTGCTCTTCCGATCTNNNNNGACTACHVGGGTATCTAATCC–3′). PCR reactions were performed in 20 μL reactions with ExTaq HS polymerase (TaKaRa BIO INC., Shiga, Japan), 0.5 μM forward and reverse primers, and 1-ng template DNA. Thermal cycling consisted of the initial denaturation at 94˚C for 2 min, followed by 20 cycles of denaturation at 94˚C for 30 s, annealing at 55˚C for 30 s, elongation at 72˚C for 30 s, and ending with samples at 72˚C for 5 min. Samples were indexed in the second PCR using index sequence inserted primers and sequenced on an Illumina MiSeq sequencer, using MiSeq Reagent Kit v3 (Illumina, CA, United States) with paired-end 300-base-pair reads. High-throughput sequencing was performed at Bioengineering Lab. Co., Ltd. (Kanagawa, Japan). Sequences were screened for chimeras using the uchime algorithm (USEARCH V8.1.1861), and putative chimeras were removed from the data set. Sequence data were processed using the Quantitative Insight into Microbial Ecology pipeline (QIIME V1.9.1). Operational taxonomic units were defined on the basis of 97% similarity clustering using QIIME with default parameters. The bacterial taxonomy assignment was performed using the Greengenes V13_8 data base. β-Diversities were also calculated on the sequence reads based on weighted and unweighted UniFrac distance matrices; subsequently, principal coordinate analysis (PCoA) was performed on the samples.

## Analysis of urinary metabolites

Midstream urine from the first-morning void was collected in a sterile screw-cap container. The urine samples were placed in a freezer immediately to avoid bacterial growth and transported after freezing for three hours or more. The urine samples were assayed by the Great Plains Laboratory, Inc. (Lenexa, KS, United States). The urinary metabolites were quantified as to their trimethylsilyl (TMS) ethers or esters, and GC-MS was performed as described in previous research [17]. Due to limitations in available data, only concentrations of 74 metabolites were reported from the spectrum analysis. All concentrations of metabolites were corrected by urinary creatinine (Cr) concentration to minimize variability of urine concentration.

## Statistical analysis

There were some cases of data loss or non-participation in the race, so each analysis used the maximum number of obtained data (at least eight participants). The number of samples used in each analysis is shown in Fig 1B. The differences in bacterial taxa characterizing the groups were evaluated by linear discriminant analysis (LDA) Effect Size (LEfSe) method with default setting on website https://huttenhower.sph.harvard.edu/galaxy/root [18]. A diagnostic check (Shapiro-Wilk test of normality) was performed prior to analysis. When normality and equal variance between sample groups were achieved, repeated measures ANOVA followed by a Bonferroni's correction or paired t-test was performed to find significant differences. For non-normally distribution data, Friedman's test with a Scheffé's multiple comparison method was performed. The difference in variance was determined using Fisher's $F$-test for α-diversity of fecal microbiota. Pearson's correlation coefficients were used to analyse associations between microbial composition and urinary metabolites. Multiple comparisons for urinary metabolites and the associations of microbial composition and urinary metabolites were adjusted using the false discovery rate (FDR) method [19]. For the difference in rate of perceived exertion as exercise load between both periods, Wilcoxon signed-rank test was conducted. Differences were considered significant at $p < 0.05$. The results of bacterial taxa were visualized as box-and-whisker plots showing: the median and the interquartile (midspread) range (boxes containing 50% of all values). Urinary metabolites were described as a mean and standard deviation.

Statistical analysis was conducted using XLSTAT v19.4 software (Addinsoft, Paris, France) and BellCurve for Excel v.3.21 (Addinsoft, Social Survey Research Information Co., Ltd., Tokyo, Japan).

# Results

## Study participants

A total of 13 Japanese college long-distance runners (nine male, four female) were enrolled, with a 4-week pre-observation period followed by a 4-week LEX-ingestion period. The baseline characteristics of the participants was as follows [shown by mean (± SD)]; height: male 173.6 (±5.1), female 158.7 (±4.9), body fat percentage: male 9.9 (±2.0), female 22.9 (±3.7), BMI: male 19.2 (±0.9), female 21.3 (±1.2). All participants ate three meals a day and had a balanced diet of common meat, fish, and vegetables, and did not intervene in their diet during the study period. Additionally, eight of 13 participants (61.5%) were taking yogurt, but during the test period, the intake was the same as usual. Two official races were held once during each period. Fig 1B shows the participant-tracking flowchart. Nine (five males and four females) of the 13 participants participated in two races. Two urine samples (two female) were not received before the start of LEX ingestion due to physical conditions not suitable for urinalysis. Of the 52 samples of fecal DNA collected, three male samples were defective, and no sequence results were obtained. As a result, the analyses of fecal microbiota (samples from five males and three females) and urinary metabolites (samples from six males and two females) were

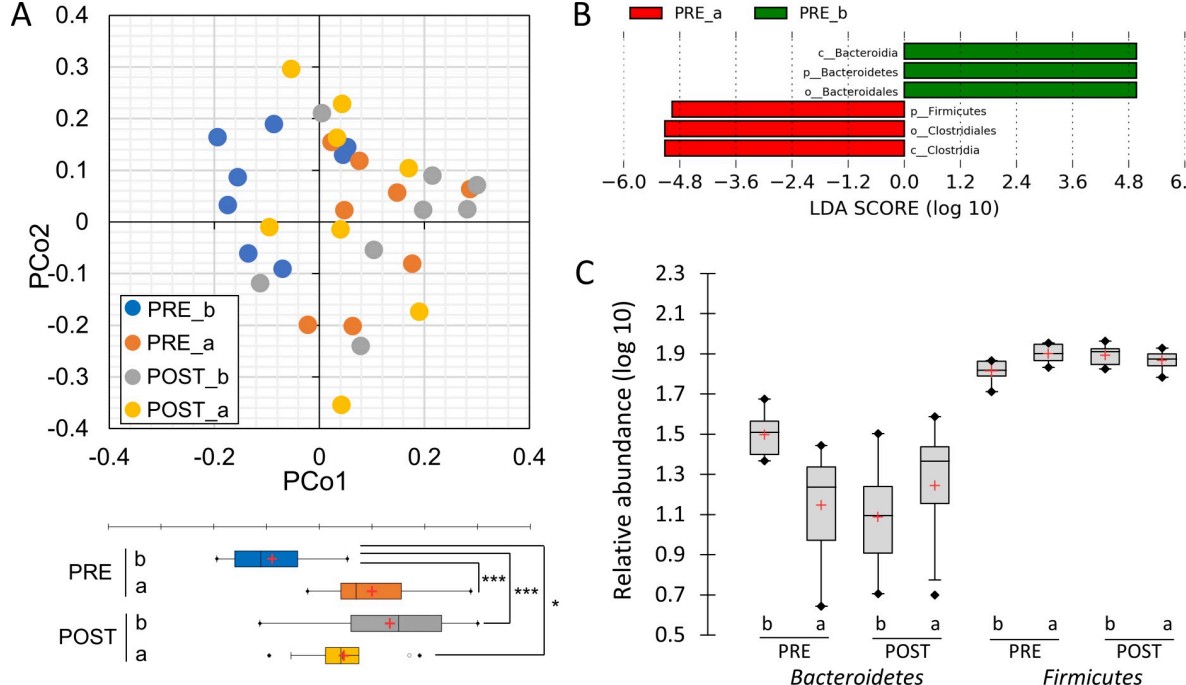

**Fig 2. Difference of fecal microbiota composition before and after the race during the pre-observation and LEX-ingestion periods.** (A) Principal coordinate analysis (PCoA) plots (weighted UniFrac analysis). (B) Linear discriminant analysis (LDA) effect size (LEfSe) analysis plot in fecal microbiota between four groups (PRE_b, PRE_a, POST_b, and POST_a). LDA scores (log10) > 2 and $p < 0.05$ are listed. (C) Histogram of the relative distribution of phyla *Bacteroidetes* and *Firmicutes*. PRE: pre-observation period. POST: LEX-ingestion period. b: before the race. a: after the race. For all box-and-whisker plots, center lines represent the median and the box edges represent the first and the third quartiles with mean (Red +). Statistical significance was determined using repeated measures ANOVA with Bonferroni's correction for PCoA plot. * and *** indicate significant differences between $p < 0.05$ and $p < 0.001$, respectively.

performed for those who participated in the two races. To analyse associations between fecal microbiota composition and urinary metabolites, data for nine participants was incorporated regardless of participation in two races.

S1 Fig shows self-reported total training time and mileage for each four-week study interval, as well as the rate of perceived exertion (RPE) for each of the two races. Athletes reported that both the training intensity and total training time were higher (approximately 25% and 17%, respectively) during the pre-observation period than during the LEX-ingestion period.

The total distance of the first race during the pre-observation period was 5,000 m for men and 3,000 m for women, and the total distance for the second race during the LEX-ingestion period was 5.4 to 12.3 km for men and 3.7 to 8.1 km for women. Thus, the RPE reported by athletes was significantly ($p = 0.048$ for the analysis of fecal microbiota, $p = 0.038$ for the analysis of urinary metabolites) higher in the second race (during the LEX-ingestion period).

## Changes in the gut microbiota composition during the test period

A total 1,115,654 (1.1 million) 16S rRNA reads were generated from fecal samples provided by eight runners, with mean of 39,095 (±7,899 SD), 30,933 (±11,006), 36,016 (±12,926), and 33,414 (±11,309) reads for PRE_b (pre-observation, before race), PRE_a (pre-observation, after race), POST_b (LEX ingestion, before race), and POST_a (LEX ingestion, after race), respectively. Reads corresponding to 12 phyla, 61 families, and 90 genera were detected in eight runners. PCoA based on weighted UniFrac distances of 16S rRNA sequences highlighted a clear differentiation of the microbial populations before and after a single race (Fig 2A). However, during the LEX-ingestion period, the changes observed in microbial populations due to race participation were small, and rather the POST_a state tended to return to the PRE_b state.

To discover significant alteration of bacterial taxa abundance in response to the race and the LEX-ingestion, we conducted LEfSe analysis. The LDA score showed higher associations of the phylum *Bacteroidetes* (including class *Bacteroidia*) and the phylum *Firmicutes* (including class *Clostridia*) with before and after the race during pre-observation period (Fig 2B). The phylum *Bacteroidetes* composition decreased and the phylum *Firmicutes* composition increased significantly after the race (Fig 2C). The *Firmicutes* to *Bacteroidetes* (F / B) ratio was significantly ($p = 0.038$) higher after the race and these trends continued even after 14 days with the LEX-ingestion period (POST_b) (S2 Fig). At the end of LEX-ingestion (POST_a), despite the increased RPE (increased effort required) reported by athletes for the race in this period, no differentially abundant features was found in the LEfSe analysis between PRE_b and POST_a. In addition, the F / B ratio at POST_a also tended to mirror those obtained during the pre-observation period (PRE-b).

Further analysis of the α-diversity of the fecal samples based on the phylum level revealed no significant differences in Chao1 and Shannon's index. However, the variance of Shannon's index before and after the race during pre-observation period was significantly different (*F*-test: $p = 0.001$) (S3 Fig). Thus, the Chao1 index represents the estimated richness of a community structure, while the Shannon's index represents both the richness and evenness of it is species diversity, indicating that the evenness of a microbial community structure is likely to fluctuate before and after the race.

## Changes in urinary metabolites due to ingestion of LEX

Seventy-three urinary metabolites were analyzed for changes in both GI microbiota and metabolism due to LEX ingestion. These compounds included fungal and yeast metabolites, bacterial metabolites, TCA cycle metabolites, neurotransmitter metabolites, and other

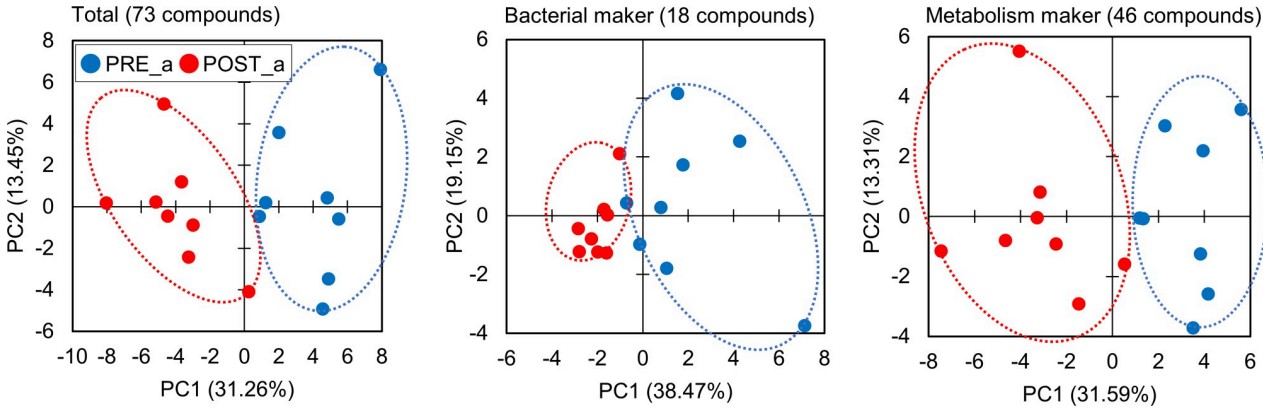

**Fig 3. Principal component analysis (PCA) score plots.** (Left panel) Total: 73 compounds. (Center panel) Bacterial marker: 18 compounds. (Right panel) Metabolism marker: 46 compounds. PRE_a: before LEX-ingestion (blue closed circle), POST_a: after LEX ingestion (red closed circle).

metabolites. Patterns of these urinary metabolites from individual samples were subjected to principal component analysis (PCA), as shown in Fig 3. Conspicuously, urinary metabolites displayed remarkable differences before and after LEX ingestion in participants who could collect urine samples and participated in two races. The differences were also clear in the cases of focusing on 18 compounds of bacterial markers and 46 compounds of metabolism markers. Additionally, it was observed that the PCA score of the bacterial markers was aggregated to a particular score after ingestion of LEX compared with the metabolic marker.

Table 1 summarizes the urinary metabolites that were significantly different before and after LEX ingestion for four weeks. There are significant changes before and after LEX ingestion in 27 of the 73 compounds. For example, arabinose is one of the yeast markers, and the urinary arabinose level before LEX ingestion was $48.25 \pm 17.17$ (SD) mmol/mol Cr, which was higher than the reference range [male ($\geq$age 13): $\leq$20 mmol/mol Cr, female ($\geq$age 13): $\leq$29 mmol/mol Cr]. However, the mean arabinose in urine after LEX ingestion decreased significantly ($p = 0.016$) to $17.00 \pm 5.21$ mmol/mol Cr, within the reference range. Other yeast and fungal markers, such as 3-oxoglutaric acid, tartaric acid, and tricarballylic acid, also had significantly reduced levels after LEX ingestion. Additionally, 3-methylglutaric acid and 3-hydroxyglutaric acid, two markers for amino acid metabolism in mitochondria, were also elevated before LEX ingestion compared to reported reference levels [3-methylglutaric acid: male ($\geq$age 13): 0.02–0.38 mmol/mol Cr, female ($\geq$age 13): $\leq$0.76 mmol/mol Cr, 3-hydroxyglutaric acid: male ($\geq$age 13): $\leq$4.6 mmol/mol Cr, female ($\geq$age 13): $\leq$6.2 mmol/mol Cr], but significant decreases were observed after LEX ingestion (3-methylglutaric acid: $p = 0.023$, 3-hydroxyglutaric acid: $p = 0.029$). Furthermore, many significant reductions were observed after LEX ingestion in the TCA cycle metabolites (2-oxoglutaric acid: $p = 0.005$, aconitic acid: $p = 0.005$) and neurotransmitter metabolites (homovanillic acid: $p = 0.006$, vanillylmandelic acid: $p = 0.006$, quinolinic acid: $p = 0.0003$, kynurenic acid: $p = 0.0004$).

## Association between gut microbiota and urinary metabolites

Subsequently, we explored the relationship between the proportion of fecal microbiota and urinary metabolite concentration. The results showed that there were several moderate correlations between microbial proportions and concentration of metabolites. Fig 4 shows a summary of these correlations. Data on tricarballylic acid and carboxylic acid, one of the fungal and yeast markers, are shown as scatter plots (Fig 5), which indicates that phylum *Firmicutes*

**Table 1. Changes in urinary metabolites before and after LEX ingestion; showing only those with significant differences.**

| Category | Component | Mean (Standard deviation) mmol/mol Creatinine | | p-value after FDR adjustment |
|---|---|---|---|---|
| | | PRE | POST | |
| Yeast and Fungi | 3-Oxoglutaric acid | 0.06 (0.05) | 0.00 (0.01) | 0.029 |
| | Tartaric acid | 0.32 (0.18) | 0.07 (0.10) | 0.049 |
| | Arabinose | 48.25 (17.17) | 17.00 (5.21) | 0.016 |
| | Tricarballylic acid | 0.11 (0.08) | 0.02 (0.05) | 0.024 |
| Bacteria | 2-Hydroxyphenylacetic acid | 0.33 (0.12) | 0.23 (0.18) | 0.037 |
| | 4-Hydroxyhippuric acid | 6.79 (3.31) | 1.94 (1.15) | 0.044 |
| Clostridia | 4-Cresol | 8.75 (7.37) | 2.76 (3.22) | 0.027 |
| TCA Cycle | 2-Oxoglutaric acid | 9.30 (5.03) | 5.68 (4.81) | 0.005 |
| | Aconitic acid | 9.78 (1.35) | 3.15 (2.10) | 0.005 |
| Mitochondrial Amino Acid | 3-Methylglutaric acid | 0.59 (0.31) | 0.21 (0.15) | 0.023 |
| | 3-Hydroxyglutaric acid | 7.59 (4.65) | 2.09 (1.62) | 0.029 |
| | 3-Methylglutaconic acid | 1.15 (0.38) | 0.73 (0.42) | 0.011 |
| Neurotransmitter | Homovanillic acid | 1.95 (0.43) | 0.81 (0.28) | 0.006 |
| | Vanillylmandelic acid | 1.41 (0.17) | 0.75 (0.36) | 0.006 |
| | Quinolinic acid | 1.58 (0.33) | 0.58 (0.28) | <0.001 |
| | Kynurenic acid | 1.18 (0.25) | 0.39 (0.25) | <0.001 |
| Pyrimidine | Thymine | 0.18 (0.05) | 0.09 (0.07) | 0.049 |
| Ketone and Fatty Acid Oxidation | Methylsuccinic acid | 1.13 (0.28) | 0.50 (0.23) | 0.004 |
| Vitamin | Pantothenic acid | 1.51 (0.53) | 0.49 (0.32) | 0.012 |
| | 3-Hydroxy-3-methylglutaric acid | 11.13 (2.54) | 4.91 (2.30) | 0.011 |
| | Methylcitric acid | 0.57 (0.18) | 0.21 (0.10) | 0.011 |
| Detoxification | Pyroglutamic acid | 21.25 (3.01) | 13.14 (4.07) | 0.011 |
| | Orotic acid | 0.31 (0.16) | 0.14 (0.12) | 0.024 |
| | 2-Hydroxyhippuric acid | 0.78 (0.68) | 0.21 (0.31) | 0.044 |
| Amino Acid | Phenylpyruvic acid | 0.85 (0.25) | 0.50 (0.43) | 0.037 |
| | 4-Hydroxyphenyllactic acid | 0.22 (0.08) | 0.06 (0.03) | 0.011 |
| Mineral | Phosphoric acid | 3.58 (0.84) [*1] | 2.67 (1.01) [*1] | 0.039 |

[*1]: mol/mol Creatinine

was positively correlated with tricarballylic acid [$r = 0.662$ ($p = 0.003$)] and carboxylic acid [$r = 0.689$ ($p = 0.002$)].

At the family level, moderate correlations were observed with several metabolites in *Porphyromonadaceae* (Figs 4 and S4). They were correlated with 3-metylglutaconic acid ($r = -0.609$, mitochondrial amino acid metabolite) kynurenic acid ($r = -0.529$, tryptophan metabolite), and pyrimidine metabolites (uracil: $r = -0.680$, thymine: $r = -0.625$). A species of *Parabacteroides distasonis* (PD, phylum *Bacteroidetes*, family *Porphyromonadaceae*) was also found to correlate with several metabolites (Figs 4 and S4). These moderate correlations included 3-methylglutaconic acid ($r = -0.630$), neurotransmitter metabolites (vanillylmandelic acid: $r = -0.602$, quinolinic acid: $r = -0.570$, and kynurenic acid: $r = -0.630$), or pyrimidine metabolites (thymine: $r = -0.554$).

To discover significant alteration of bacterial taxa abundance in response to LEX ingestion, we conducted LEfSe analysis for 11 participants regardless of whether they participated in the two races. As a result, the LDA scores showed significant bacterial differences before and after LEX ingestion (S5 Fig). The composition of *Odoribacter* (phylum *Bacteroidetes*, family

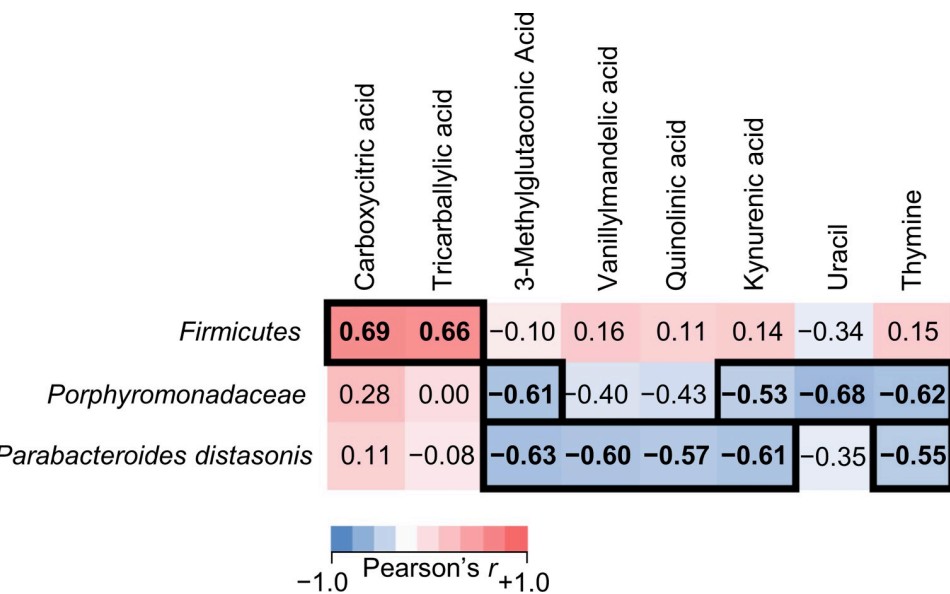

**Fig 4. Pearson's correlation heat map correlations between fecal microbiota and urinary metabolites.** The color strip (down legend) represents the correlation between each item. The darker color indicates that the correlation coefficient increases gradually. For example, red shows a positive correlation, while blue shows a negative correlation. The values indicate Pearson's correlation coefficients ($r$), and bold letters indicate $p < 0.05$.

*Odoribacteraceae*), *Turicibacter* (phylum *Firmicutes*, family *Turicibacteraceae*), and PD increased, and *Oribacterium* (phylum *Firmicutes*, family *Lachnospiraceae*) composition decreased after LEX ingestion. PD was the only biomarker detected at species level and has the highest abundance ratio among these biomarkers.

## Discussion

This study provides two insights into the competition and supplement ingestion on endurance athletes as a pilot study. To the best of our knowledge, this is the first study combining the

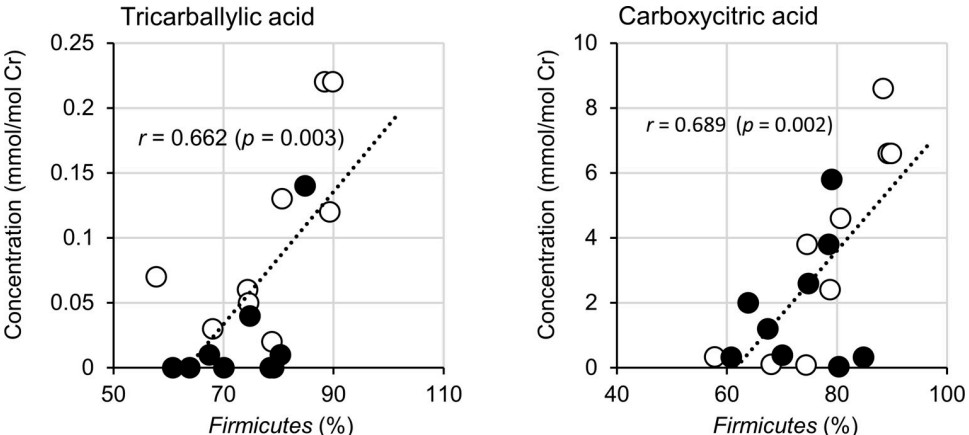

**Fig 5. Correlations between urinary metabolites and bacterial composition at the phylum level.** Phylum *Firmicutes* versus tricarballylic acid (A) and carboxylic acid (B). Open and closed circles represent values in the pre-observation period (PRE) and LEX-ingestion period (POST), respectively. Pearson's correlation coefficients ($r$) are shown for each plot with $p$-values. The analysis was conducted using data from 18 points from two samples given by nine participants.

analysis of changes in the gut microbiota and urinary metabolites of endurance athletes, with and without LEX supplementation. This study demonstrated that athletes who competed had changes in the composition of the gut microbiota. It was also suggested that LEX ingestion suppresses yeast and fungal overgrowth and improves mitochondrial metabolism, including the TCA cycle.

There are several limitations to this study. First, we acknowledge limitation to this study because it lacked a control group, and that the placebo effect may have explained the observed effects of LEX ingestion. Also, it was not possible to distinguish whether changes in microbiota on LEX or training with the used study design. Additionally, we adjusted the schedule so that participants could compete in two races during the research period. Still, it was challenging to align the competition intensity with the observation period. Second, there is a lack of physiological data on exercise and clinical data on the GI tract. Third, an in-depth analysis of the participants' diets was not provided in this study due to various factors. The diet and exercise are the two factors that significantly influence the gut microbiota [4, 5]. Therefore, a randomized placebo-controlled study needs to be conducted with data on their physiological data on exercise and diet to analyze the role of diet, including supplements and exercise in influencing the gut microbiota taxonomic composition in athletes.

In this study, fluctuations in phylum-level composition of the gut microbiota were observed during the pre-observation (non-supplementation) period, with a decrease in *Bacteroidetes* and an increase in *Firmicutes* before and after the race. It is important to note that samples were collected on the day before and six days after the race, but the trend continued for at least 14 days. Several studies in animal models and humans have found correlations between specific alterations in the gut microbial community structure and exercise, although these were studies of the effects of long-term exercise. Most of the published studies on mouse models have examined the combined effects of exercise, dietary interventions, and diseases. Choi et al. (2013) reported that mice using a running wheel exercise method had an increase in phylum *Firmicutes* but decreases in phyla *Tenericutes* and *Bacteroidetes*, which attenuated the changes in gut microbiota induced by oral exposure to polychlorinated biphenyls (PCB) [20]. Similarly, in both type-two diabetic and control mice, exercise resulted in a greater abundance of *Firmicutes* species and lower *Bacteroides/Prevotella* genera than sedentary mice [21]. In contrast, Evans et al. (2014) described increases in the phylum *Bacteroidetes* and decreases in the phylum *Firmicutes* in a manner that was proportional to the distance ran by mice fed a high-fat diet [22]. In a short-term human study, Zhao et al. (2018) reported that a half-marathon race induced increasing richness in the families *Coriobacteriaceae* and *Succinivibrionaceae*, which belong to the phyla *Actinobacteria* and *Proteobacteria*, respectively [23]. Therefore, it is thought that exercise affects gut microbiota composition and can fluctuate due to a single competition. The difference in the dispersion of the Shannon's index before and after the race suggests that the bacterial composition is relatively unstable after the race. Participants in this study undergo daily training, and changes in the gut microbiota at the phylum level may be due to the rigorous physical demands of an official race and the psychological effects of the competition. It is interesting that this change continues for a relatively long period, and further research is needed to delve into how the microbiome is related to the conditioning of athletes.

Long-distance runners need sustained energy for a long time of continuous exercise and their enduring workouts. The metabolic demands for skeletal muscle, liver, kidneys, and adipose tissue are increased within a day or a few days after running. Studies of the correlation between microbiota composition and food intake have reported many correlations. Fat and energy intake were the largest connected components in the network between bacterial taxa [23]. Changes in the bacterial composition may be induced to regulate energy and hormonal balance, or to response to changes in the diet ingested. However, athletes are doing high load

training on a daily basis. Considering this, the psychological impact of competition may also be related to the changes in the bacterial composition in a single competition. Also, these changes may have major impact on maintaining the athlete's post-competition condition. Therefore, it is necessary to conduct the study while accurately grasping the dietary intake and mental status before and after the competition.

The change in microbial composition before and after the race during the LEX-ingestion period were smaller than that seen during the pre-observation period, as the bacterial composition on the final day of LEX ingestion (POST_a) was similar to the initial state (PRE_b). However, directly comparing the two study periods is difficult because sampling after the race was longer (6 days) during the LEX-ingestion period. Moreover, it is possible that not enough time elapsed between the pre-observation (pre-ingestion) period and the LEX-ingestion period to allow the microbiome to return to baseline, which would potentially confound the results seen. In addition, it is necessary to consider that the exercise intensity may differ slightly depending on the observation period, potentially impacting the microbiota. However, because the change in the POST_b microbiota is not maintained (i.e., reverts to the original pre-race microbial composition), we expect that LEX ingestion will have the ability to suppress changes in the gut microbial resulting from the participation in an official competition.

We also investigated changes in urinary metabolites before and after the ingestion of LEX in this study. Unfortunately, it is necessary to refrain from consuming sports drinks and fruits that affect urine metabolites from the day before the urine collection and considering the maintenance of the condition before the race. The analysis of urinary metabolites on the day before the competition could not be carried out at the same time as the fecal collection. Before LEX ingestion, the arabinose concentration (48.25 mmol/mol Cr on mean) is considerably higher than the reference range. Arabinose is a marker for *Candida* species (yeast), and increasing arabinose levels in urine correlate with yeast overgrowth [24]. After ingestion of LEX, urinary arabinose levels significantly decreased. Additionally, as yeast and fungal markers, significant reductions in 3-oxoglutaric acid, tartaric acid, and tricarballylic acid were also observed [25–27]. Generally, these results suggest that continuous ingestion of LEX suppresses yeast and fungal overgrowth.

In humans, fungi colonize the gut shortly after birth [28]. Human fecal gut mycobiome is low in diversity compared to its bacterial composition, and dominated by yeast including *Saccharomyces*, *Malassezia*, and *Candida* [29]. The relationship between diet and fungi has also been studied, and *Candida* is abundant in relation to recent carbohydrate consumption [30]. The presence of fungi is associated with the exacerbation of several human diseases, including inflammatory bowel disease and colorectal cancer [31–33]. It has also been reported that the growth of fungi in the intestine is involved in the inflammatory response and enhances the allergic response [34]. Endurance athletes routinely perform long-distance running training (for the participants of this study, the monthly mean of running distance was about 260km), and this excessive exercise load may reduce the momentum of gut microbiota, resulting in the growth of yeast and fungi. The overgrowth of fungi in the gut is involved in facilitating systemic inflammation and increasing fatigue, which affecting the maintenance of an athlete's physical condition. Ingestion of LEX may suppress the excessive growth of fungi and may be effective for daily conditioning.

Of the nine markers related to bacterial growth, three showed a significant decrease after LEX ingestion. These metabolites are tyrosine and phenylalanine metabolites, and these results suggest that the overgrowth of specific bacteria, including *Clostridia* and microbial dysbiosis is suppressed. Furthermore, changes in urinary metabolites reflecting neurotransmitter levels were observed in relation to the metabolism of phenylalanine, tyrosine, or tryptophan. We recently reported that LEX ingestion decreases urinary indoxyl sulfate, a tryptophan

metabolite derived from the metabolism of gut microbiota [12]. Changes in these metabolites are also presumed to be mediated by the gut microbiota.

Another notable change in urinary metabolites before and after LEX ingestion are markers for mitochondrial metabolites, including the TCA cycle. Of the nine compounds in these markers, the values before LEX ingestion were within the standard range, but five compounds showed a significant decrease after LEX ingestion. We hypothesize that continuous ingestion of LEX improves mitochondrial metabolism, leading to the maintenance of energy acquisition and substrate availability for assimilation processes such as lipogenesis. 3-Methylglutaric acid and 3-methylglutaconic acid are leucine metabolites, and leucine is known as a potent stimulator of protein synthesis by stimulating the mammalian target of rapamycin (mTOR) [35]. Considering these points, the maintenance of GI environment may enhance recovery from fatigue and muscle repair by modulating inflammation in daily training.

Vitamin B contributes to the process of the TCA cycle as cofactors/enzymes such as FAD (B2) and NAD (B3), as components of CoA (B5), or as coenzyme Q10 (B5) [36]. In relation to vitamins, significant changes were observed in pantothenic acid (B5), 3-hydroxy-3-methylglutaric acid (precursor of coenzyme Q10), and methylcitric acid (an indicator of biotin). Although dietary B vitamins are absorbed through the small intestine, they can also be supplied by the gut microbiota biosynthesis. Generally, it is hypothesized that the activation of mitochondrial metabolism and TCA cycle by LEX ingestion may be mediated through gut microbiota.

We then analyzed the relationship between the composition of gut microbiota and urinary metabolites. Correlations were detected between phylum *Firmicutes*, and yeast and fungal markers, including tricarballylic acid and carboxycitric acid, positively correlated. It is hypothesized that the imbalance of the gut microbiome may induce excessive growth of yeast and fungi. This has been demonstrated previously. For example, antibiotics treatment in mice leads to major fungi expansions [34, 37], suggesting that the balance of bacterial composition controls the prevalence of the fungus in the gut. In the composition of human gut microbiota phyla *Firmicutes* and *Bacteroidetes* are the two major dominant [38]. The *Firmicutes* to *Bacteroidetes* ratio has been extensively examined for human and mouse gut microbiota. Multiple studies reveal that the F/B ratio is correlated with obesity and other diseases [39–42]. It is also presumed that the balance between phyla *Firmicutes* and *Bacteroidetes* is involved in controlling fungal occupancy in athletes.

PD is the only bacterial species found to correlate with urinary metabolites at the species level. The genus *Parabacteroides*, including PD are a gram-negative anaerobe and defined as one of the 18 core members of the gut microbiota of humans [43]; thus, it is thought to be involved in important physiological functions in the host. The abundance of PD is relatively lower in patients with obesity, nonalcoholic fatty liver disease (NAFLD), inflammatory bowel disease (IBD), and multiple sclerosis [44–47]. Moreover, treatment with live PD in mice was shown to have anti-inflammatory effects, reduce weight gain, improve glucose homeostasis, correct obesity-related abnormalities, and induce regulatory T lymphocytes from naïve CD4[+] T cells [48, 49]. It has also been suggested that these effects are regulated by PD via succinic acid and secondary bile acid production or suppression of TLR4 and AKT signals [49]. These findings support the gut PD as a promising symbiont that can modulate host metabolism to potentially alleviate metabolic dysfunction. In this study, many urinary metabolites were correlated with PD composition, suggesting associations to leucine, phenylalanine, tyrosine, and tryptophan metabolism. As a constituent of gut microbiota, PD may also be an important marker for athlete conditioning as it is also associated with anti-inflammatory responses. For reference, changes in PD composition due to LEX ingestion significantly increase in comparison before and after LEX ingestion regardless of whether they participated in two races.

Although PD administration has been suggested as a probiotic, it is important to identify food ingredients that regulate endogenous PD rather than supplemental PD.

Conclusively, this study provides insight into the impact of competition and health effects of oral ingestion of LEX of endurance athletes, focusing on their microbiome composition. Our data reveal that a single, official endurance race can rapidly induce striking compositional changes in gut microbiota. These alterations can also potentially last for a relatively long period. Analysis of urinary metabolites suggested that the participants may also be experiencing yeast and fungal overgrowth in the gastrointestinal system. Therefore, LEX ingestion may also have an overall beneficial effect on this aspect as well. These data suggested that LEX ingestion may also improve mitochondrial metabolism and amino acid metabolism and may effectively maintain the daily metabolic homeostatic condition of athletes. A relationship between fungal growth and occupancy of the phylum *Firmicutes* was also observed, and PD, which is linked with many urinary metabolites, may also be an important indicator for athlete condition. Further studies, however, must elucidate the impact of competitive events on elite athletes' gut microbiome and explore options for mitigating adverse effects on the GI environment, such as in the case of LEX administration.

## Supporting information

**S1 Fig. Exercise loads in the pre-observation period and the LEX-ingestion period.** Total mileage, training time, and rate of perceived exertion (RPE) are shown. (A) Exercise load of 8 participants in the analysis of fecal microbiota. n = 8 (5 male /3 female). (B) Exercise load of 8 participants in the analysis of urinary metabolites. n = 8 (6 male/2 female). PRE: pre-observation period. POST: LEX-ingestion period. Values represent scatter plot with median (black line) and mean (Red +). Statistical significance was determined using a paired *t*-test for millage and training time. Statistical significance was determined using the Wilcoxon signed-rank test for RPE. Significant differences between PRE and POST: *$p < 0.05$.
(JPG)

**S2 Fig. Changes in *Firmicutes* to *Bacteroidetes* ratio of fecal microbiota.** POST: LEX-ingestion period. b: Before the race. a: After the race. Values represent box-and-whisker plots with mean (Red +). Statistical significance was determined using Friedman's test with a Scheffé's multiple comparison method. * indicate a significant difference of $p < 0.05$.
(JPG)

**S3 Fig. Changes in α-diversity of fecal microbiota at the phylum level.** PRE: Pre-observation period. POST: LEX-ingestion period. b: Before the race. a: after the race. Values represent box-and-whisker plots with mean (Red +). Statistical significance was determined using repeated measures ANOVA with Bonferroni's correction. The difference in variance was determined using Fisher's *F*-test. †† indicate a different variance between the two samples: $p < 0.01$.
(JPG)

**S4 Fig. Correlation plots of fecal bacterial composition and urinary metabolites.** The results of *Porphyromonadaceae* and *Parabacteroides distasonis* are shown. Open and closed circles represent values on the pre-observation period (PRE) and LEX-ingestion period (POST), respectively. Pearson's correlation coefficients (*r*) are shown for each plot with *p*-values. The analysis was performed using data from 18 points of nine participants.
(JPG)

**S5 Fig. Differential abundance of microbial taxa before and after LEX ingestion.** (A) Linear discriminant analysis (LDA) effect size (LEfSe) analysis plot of taxonomic biomarkers in fecal

microbiota between PRE_a and POST_a. LDA scores (log10) > 2 and $p < 0.05$ are listed. (B) Histogram of the relative distribution of *Parabacteroides distasonis*. Values represent box-and-whisker plots with mean (Red +). The analysis included 11 participants who had data for fecal microbiota before and after LEX ingestion.
(JPG)

## Acknowledgments

We would like to thank all the participants who took part in the study. Additionally, we appreciate California Nutrients, Inc. and Bioengineering Lab. Co., Ltd. for their technical assistance in conducting urine analysis and high-throughput sequencing, respectively. The English in this document has been vetted by a professional service (Enago) and an editor who is a native English speaker.

## Author Contributions

**Conceptualization:** Masaaki Sugita, Keisuke Yonekura, Yasuhiro Sasuga.

**Data curation:** Mina Fukuchi, Makoto Banjo.

**Formal analysis:** Mina Fukuchi, Yasuhiro Sasuga.

**Funding acquisition:** Masaaki Sugita, Keisuke Yonekura, Yasuhiro Sasuga.

**Investigation:** Mina Fukuchi, Makoto Banjo.

**Methodology:** Masaaki Sugita, Yasuhiro Sasuga.

**Project administration:** Masaaki Sugita, Yasuhiro Sasuga.

**Resources:** Mina Fukuchi, Masaaki Sugita, Makoto Banjo, Keisuke Yonekura, Yasuhiro Sasuga.

**Supervision:** Masaaki Sugita, Yasuhiro Sasuga.

**Validation:** Mina Fukuchi, Masaaki Sugita, Yasuhiro Sasuga.

**Visualization:** Mina Fukuchi, Yasuhiro Sasuga.

**Writing – original draft:** Mina Fukuchi, Yasuhiro Sasuga.

**Writing – review & editing:** Masaaki Sugita, Makoto Banjo, Keisuke Yonekura, Yasuhiro Sasuga.

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
