## [Decision Letter · Decision Letter 0]

24 Jun 2021

PONE-D-21-11683

The impact of a competitive event and the efficacy of a lactic acid bacteria-fermented soymilk extract on the gut microbiota and urinary metabolites of endurance athletes: an open-label pilot study

PLOS ONE

Dear Dr. Sasuga,

Thank you for submitting your manuscript to PLOS ONE. After careful consideration, we feel that it has merit but does not fully meet PLOS ONE’s publication criteria as it currently stands. Therefore, we invite you to submit a revised version of the manuscript that addresses the points raised during the review process.

We look forward to receiving your revised manuscript.

Kind regards,

Krzysztof Durkalec-Michalski, Ph.D

Academic Editor

PLOS ONE

Journal Requirements:

2. In the competing interests statement within the manuscript and in the online submission form, please declare your affiliation with B&S Corporation and thoroughly report any potential competing interests related to this affiliation. For example, the product used in your study, LACTIS, is produced by B&S Corporation - this should be clearly stated in your competing interest statement. Thank you for your attention to this request.

Reviewers' comments:

Reviewer's Responses to Questions

**Comments to the Author**

1. Is the manuscript technically sound, and do the data support the conclusions?

Reviewer #1: No

Reviewer #2: Yes

2. Has the statistical analysis been performed appropriately and rigorously? 

Reviewer #1: I Don't Know

Reviewer #2: No

3. Have the authors made all data underlying the findings in their manuscript fully available?

Reviewer #1: Yes

Reviewer #2: Yes

4. Is the manuscript presented in an intelligible fashion and written in standard English?

Reviewer #1: Yes

Reviewer #2: Yes

5. Review Comments to the Author

Reviewer #1: The authors presented a very interesting pilot study in which they evaluated the effects of lactic acid bacteria-fermented soymilk extract on the microbiome and metabolome of athletes before and after exercise during competition. The researchers made several interesting observations; however, the conclusions they drew are not appropriate. The most important conclusions are that the primary and secondary aims and research hypotheses should be precisely defined; a randomized placebo-controlled study should be conducted (comparator without bacteria); the sample size should be calculated before the study (in the pilot study, authors obtained some data, which will be helpful to do it). Moreover, physiological data on exercise and clinical data on the gastrointestinal tract must be included in the analysis, and it is necessary to record the diet of athletes. The discussion should be rewritten in this direction. In addition, several comments on the manuscript should be included:

line 60: Please cite the following position: Kulecka M et al. The composition and richness of the gut microbiota differentiate the top Polish endurance athletes from sedentary controls. Gut Microbes. 2020 Sep 2;11(5):1374-1384.

Line 69: Not only the GI tract is the target of probiotics; please inform readers about other benefits for athletes receiving probiotics.

Line 71: Probiotics activity is strain-specific. In the study referenced 11, the list of strains is not shown; tehrefore, there are fermenting bacteria, not probiotics.

Line 95: What do authors mean: "improving GI environment", we do not know the healthy microbiota and only speculate about it.

Line 109: With the used study design, it is impossible to distinguish whether the microbiota changes depended on LEX or training; this issue must be clearly stated in the discussion.

Line 120: The composition of the product, the names of the strains and their quantities were not given.

Line 144: Why was urine not tested at the same time points as a stool?

Results: How the authors related the compositionality problem?

Reviewer #2: In the work The impact of a competitive event and the efficacy of a lactic acid bacteria-fermented soymilk extract on the gut microbiota and urinary metabolites of endurance athletes: an open-label pilot study Fukuchi et al. describe the effects of supplement LEX on fecal bacteriome and urine metabolome of endurance athletes in competitive environment. The study itself seems well-designed but its description, especially the Methods section could be improved upon.

Major remarks:

1. Abstract

“When the fecal microbiota was investigated before and after a race without the use of a supplement (pre-observation period), there was an increase in the abundance of phylum Bacteroidetes and decrease in the abundance of Firmicutes.” The opposite is mentioned in the results section as well as in Figure 2B

2. Methods

a. General – statistical methods are poorly described – some of them, like exclusion of samples by Smirnov-Grubbs test (line 591) are only mentioned in results and/or figure description.

b. Line 178 – software versions for usearch and QIIME should be given

c. Line 181 – which OTU picking strategy from QIIME was used? What database was used to assign taxonomic classification?

d. Line 201 – no post-hoc test is identified for Friedman’s test

e. Line 202 – no mention of multiple comparison adjustment of p-value is given here yet it is present in the results

f. Line 204 –Friedman’s test (a non-parametric method) was used previously for differential analysis of microbial taxa which indicates the authors believe the taxa abundance distribution deviates from normal distribution. But here, Pearson’s correlation coefficient is computed which is best suited for variables with normal distribution. Could authors justify their choice? Wouldn’t Spearman correlation coefficient be better?

g. Line 205 and 201 – two different p-value adjustment strategies are used: FDR and Bonferroni. Could authors justify their choice here?

3. Results

a. It would be good to see some baseline characteristics of athletes other than age, such as their BMI. Information on diet is also missing – even general remarks (was anyone on vegan/vegetarian diet? Was any other supplementation received?) would be helpful if such information is available.

Minor remarks:

Some wording choice is unusual, such as Bonferroni’s correction coefficient (line 260). It would be better to say simply Bonferroni’s correction of Bonferroni’s p-value adjustement

6. PLOS authors have the option to publish the peer review history of their article (what does this mean?). If published, this will include your full peer review and any attached files.

Reviewer #1: No

Reviewer #2: No

---

## [Author Response · Author response to Decision Letter 0]

8 Jul 2021

Krzysztof Durkalec-Michalski, Ph.D

Academic Editor

PLOS ONE 

Yasuhiro Sasuga, Ph.D

Hachioji Center for Research & Development

B&S Corporation Co., Ltd.

8th July 2021

Dear Durkalec-Michalski

Response to reviewers’ comments [PONE-D-21-11683]

Thank you for reviewing our manuscript and for the helpful comments provided.

Please find enclosed our response to the comments raised. The line numbers provided here are based on our revised manuscript with tracked changes.

1. Reviewer 1

■ The authors presented a very interesting pilot study in which they evaluated the effects of lactic acid bacteria-fermented soymilk extract on the microbiome and metabolome of athletes before and after exercise during competition. The researchers made several interesting observations; however, the conclusions they drew are not appropriate. The most important conclusions are that the primary and secondary aims and research hypotheses should be precisely defined; a randomized placebo-controlled study should be conducted (comparator without bacteria); the sample size should be calculated before the study (in the pilot study, authors obtained some data, which will be helpful to do it). Moreover, physiological data on exercise and clinical data on the gastrointestinal tract must be included in the analysis, and it is necessary to record the diet of athletes. The discussion should be rewritten in this direction. 

We appreciate the reviewer’s comment. We have divided the conclusions into both primary and secondary. We reflected that in abstracts (line 46–51) and discussion (line 418–425). In the discussion section, we added the limitation of this study and stated that it is necessary to monitor physiological data on exercise and clinical data on the GI tract and/or diet (line426-440). Also, we have noted that a randomized placebo-controlled study should be conducted in the future (line 440–443).

■ In addition, several comments on the manuscript should be included:

line 60: Please cite the following position: Kulecka M et al. The composition and richness of the gut microbiota differentiate the top Polish endurance athletes from sedentary controls. Gut Microbes. 2020 Sep 2;11(5):1374-1384.

We accepted this reviewer’s suggestion. Reference 8 (line 73 and line 738–743). 

■ Line 69: Not only the GI tract is the target of probiotics; please inform readers about other benefits for athletes receiving probiotics.

We accepted this reviewer’s suggestion. We added some key words (stress, host immunity, upper respiratory tract infections). Line 78–80.

■ Line 71: Probiotics activity is strain-specific. In the study referenced 11, the list of strains is not shown; tehrefore, there are fermenting bacteria, not probiotics.

We accepted this reviewer’s suggestion. We have inserted the word “fermented bacterial products” in the previous sentence (line 81). 

■ Line 95: What do authors mean: "improving GI environment", we do not know the healthy microbiota and only speculate about it.

We accepted this reviewer’s suggestion. Certainly no one knows a healthy microbiota. Therefore, it was rewritten to affect the GI environment (line 108).

■ Line 109: With the used study design, it is impossible to distinguish whether the microbiota changes depended on LEX or training; this issue must be clearly stated in the discussion.

We accepted this reviewer’s suggestion. We have described the discussion section (line 432–434). It also clearly stated that a randomized, placebo-controlled trial should be conducted (line 440–443).

■ Line 120: The composition of the product, the names of the strains and their quantities were not given.

Since the test product is a commercial product and does not contain live bacteria, we did not mention the name of the strains, but we added the number of bacteria at the time of fermentation, the extraction process, and the expected bacterial equivalent of the extract, according to the opinion of reviewer’s suggestion (line 137–140).

■ Line 144: Why was urine not tested at the same time points as a stool?

It is necessary to refrain from consuming sports drinks and fruits that affect urine metabolites from the day before the urine collection. Therefore, considering the maintenance of the condition before the race, the urine collection on the day before the race could not be carried out at the same time as the fecal collection. We have described the discussion section (line 510–514).

■ Results: How the authors related the compositionality problem?

Thank you for pointing out the viewpoint. We think that the gut bacterial composition (including its metabolites) and dietary intake are closely related. This study did not investigate accurate dietary intake (especially before and after the race). Fat and energy have been reported to be the largest connected components in the network between bacterial taxa. Changes in the bacterial composition may be induced to regulate energy and hormonal balance, or to respond to changes in the diet ingested. It is also thought that the mental stress of competition is also involved. In any case, the changes in the bacterial composition due to a single competition may have major impact on maintaining the athlete's post-competition condition. These perspectives have been added to the discussion section (line 480–493).

2. Reviewer 2

■ 1. Abstract

“When the fecal microbiota was investigated before and after a race without the use of a supplement (pre-observation period), there was an increase in the abundance of phylum Bacteroidetes and decrease in the abundance of Firmicutes.” The opposite is mentioned in the results section as well as in Figure 2B.

We appreciate the reviewer’s comment. We collected the text (line 36 and 37).

■　2. Methods

a. General – statistical methods are poorly described – some of them, like exclusion of samples by Smirnov-Grubbs test (line 591) are only mentioned in results and/or figure description.

We appreciate the reviewer’s comment. We have added the description of statistical analysis (line 219–221). Also, since there was a lack of description of statistical methods for the item of exercise load, we added it (line 232–236).

b. Line 178 – software versions for usearch and QIIME should be given

We appreciate the reviewer’s comment. We have described the software versions for usearch and QIIME (line 194 and 196).

c. Line 181 – which OTU picking strategy from QIIME was used? What database was used to assign taxonomic classification?

We appreciate the reviewer’s comment. We used Greengenes 13_8 database, so, we have added it to the text (line 199).

d. Line 201 – no post-hoc test is identified for Friedman’s test

e. Line 202 – no mention of multiple comparison adjustment of p-value is given here yet it is present in the results

We appreciate the reviewer’s comment. We have described post-hoc test for Friedman’s test. We used a Scheffé's method　for Friedman’s test, as post-hoc test (line 224).

f. Line 204 –Friedman’s test (a non-parametric method) was used previously for differential analysis of microbial taxa which indicates the authors believe the taxa abundance distribution deviates from normal distribution. But here, Pearson’s correlation coefficient is computed which is best suited for variables with normal distribution. Could authors justify their choice? Wouldn’t Spearman correlation coefficient be better?

We appreciate the reviewer’s comment for statistical processing. Appropriate methods were not applied for some items for differential analysis of microbial taxa. The taxa abundance distribution of the phylum Bacteroidetes and Firmicutes were normal distribution. Therefore, we have changed to one-way ANOVA with Bonferroni’s correction (line 221–223, line 296–298). As for the F / B ratio, the normality was not recognized, so the Friedman ’s test was used (line 223–224, line298–299). Similarly, PCoA analysis confirmed normality, so we have changed to one-way ANOVA. Because the bacterial distribution was normal, the analysis of the relationship between urinary metabolites and bacterial composition remained Pearson’s correlation coefficient. Bacterial taxa and diversity analysis of variance in S2 Fig and S3 Fig were also normal, so we changed to ANOVA (line 651 and 659). We have changed some statistical analysis methods, but the results are not affected. Due to these changes, Fig 2(A and B), S2 Fig and S3 Fig have been replaced.

g. Line 205 and 201 – two different p-value adjustment strategies are used: FDR and Bonferroni. Could authors justify their choice here?

We appreciate the reviewer’s comment. If the number of multiple comparisons in the test was five or less, Bonferroni’s correction was used, and more than that, FDR was used. So, FDR was used to analyze the urinary metabolites and the relationship of bacterial composition and urinary metabolites, which have many items to be compared. Since the description was halfway, we have added it to the text (line 229–231).

■ M&M: following the reagents, the information of company, city, country should be included.

It describes the reagents we used. Also, regarding the contract examination, the company is described.

■　3. Results

a. It would be good to see some baseline characteristics of athletes other than age, such as their BMI. Information on diet is also missing – even general remarks (was anyone on vegan/vegetarian diet? Was any other supplementation received?) would be helpful if such information is available.

We appreciate the reviewer’s comment. We added a paragraph describing the background of the participants (line 247–253).

■ Minor remarks:

Some wording choice is unusual, such as Bonferroni’s correction coefficient (line 260). It would be better to say simply Bonferroni’s correction of Bonferroni’s p-value adjustement.

We accepted this reviewer’s suggestion (line 297).

3. Additional changes

(i) Changed the notation of Co-author affiliation. The affiliation at the time of the study and the current affiliation were separated.

(ii) We have an English proofreader correct the detailed English wording and change it. See revised manuscript with tracked changes.

We hope that we have addressed all the issues raised, and would be happy to clarify further on any other issues. Thank you for consideration of our manuscript for publication in your journal.

Yours sincerely,

Yasuhiro Sasuga

---

## [Decision Letter · Decision Letter 1]

1 Nov 2021

PONE-D-21-11683R1The impact of a competitive event and the efficacy of a lactic acid bacteria-fermented soymilk extract on the gut microbiota and urinary metabolites of endurance athletes: an open-label pilot studyPLOS ONE

Dear Dr. Sasuga,

Thank you for resubmitting your manuscript to PLOS ONE and making your manuscript significantly improved. After careful consideration, we feel that it has merit but does not fully meet PLOS ONE’s publication criteria as it currently stands. The work requires additional compliance with the recommendations of the reviewer and the editor's comments. Therefore, we invite you to submit a revised version of the manuscript that addresses the points raised during the review process. 

We look forward to receiving your revised manuscript.

Kind regards,

Krzysztof Durkalec-Michalski, Ph.D

Academic Editor

PLOS ONE

Journal Requirements:

Additional Editor Comments (Please be aware that comments are referenced to text with the track changes option):

Line 27 – correct “resent” to “recent”.

Line 66 – try to reduce even one "and" in this sentence - is repeated too often.

Line 133 - The heading "TEST ARTICLE" is unclear. Consider Improving - Maybe "Supplementation". Likewise, in the later sections (e.g. lines 147-148) - the "test article" is misleading.

Lines 151-153 - Please add the necessary clarifications what exactly was and how "An official race" was conducted.

Line 164 – unclear part – authors wrote “and exercise intensity in order to evaluate exercise intensity” – do the authors mean “and exercise specificity in order to evaluate exercise intensity”? Please revise it.

Lines 166-167  – unclear part – authors wrote “ exercise load of the race when participating in the race”. Maybe some of the terms "race" can be replaced with "exercise tests" or "competition".

The statements in the methodical part of biological material storage "as soon as possible"  (lines 159 and 175) and are quite unstable and unclear - what does that mean? What was the criterion or the range of time?

Lines 216-241 - In the description of the statistical parts it should also be clearly indicated whether the minimum sample size has been achieved, bearing in mind some losses.

Line 354 - put "concentration" behind "arabinose".

Line 421 - the authors write "an official competition" - but what was the detailed specificity of these efforts. Were they real competitions in which competitors competed with other competitors (not participating in the project) or simulated competitions in which efforts imitating competitions were carried out?

Line 477 - try to use a more formal language and replace the word "fascinating".

Lines 480-481 - the authors wrote "During long-distance running, the total energy of the body is very consumed" - this sentence is confusing. Do the authors mean that the body's energy expenditure is significant / elevated?

Line 484 - You should probably put "intake" behind “Fat and energy”

Line 534 – change “is about 260km” to past tense.

Lines 570-571 - the sentence, however, may be too optimistic despite the assurance that this is a hypothetical opinion of the authors. One should approach this even more carefully and point to "possible" or "potential" activation and that LEX ingestion "may be" mediated.

Reviewers' comments:

Reviewer's Responses to Questions

**Comments to the Author**

1. If the authors have adequately addressed your comments raised in a previous round of review and you feel that this manuscript is now acceptable for publication, you may indicate that here to bypass the “Comments to the Author” section, enter your conflict of interest statement in the “Confidential to Editor” section, and submit your "Accept" recommendation.

Reviewer #1: (No Response)

Reviewer #2: All comments have been addressed

2. Is the manuscript technically sound, and do the data support the conclusions?

Reviewer #1: Yes

Reviewer #2: Yes

3. Has the statistical analysis been performed appropriately and rigorously? 

Reviewer #1: Yes

Reviewer #2: Yes

4. Have the authors made all data underlying the findings in their manuscript fully available?

Reviewer #1: Yes

Reviewer #2: Yes

5. Is the manuscript presented in an intelligible fashion and written in standard English?

Reviewer #1: Yes

Reviewer #2: Yes

6. Review Comments to the Author

Reviewer #1: Most of the comments have been taken into account by the authors. However, I maintain my opinion that the article should include a list of the strains included in the product tested. As shown for example in the work of van Baarlen P et al. (van Baarlen P et al. Differential NF-kappaB pathways induction by Lactobacillus plantarum in the duodenum of healthy humans correlating with immune tolerance. Proc Natl Acad Sci U S A. 2009;106(7):2371-6.) killed strains have their specific physiological activities and it can be suspected that it is strain dependent.

Authors should clarify the answer to the compositionality problem. Microbiome data are compositional (relative) and give no information about absolute abundances, regardless of normalization procedures (Gloor et al., 2017). Therefore relevant statistical methods, e.g., based on log-ratios must be used, to avoid false-positive results (Knight et al., 2018; Mandal et al.,2015; Weiss et al., 2017). Information on the increase in abundance of a particular bacterial genus must be linked to the identification of the reference point (i.e. another genus) against which this occurred (Christensen et al., 2009;Morton et al., 2019, 2017). (Christensen et al., 2009;Morton et al., 2019, 2017).

References:

Christensen, K., Doblhammer, G., Rau, R., Vaupel, J.W., 2009. Ageing populations: the challenges ahead. Lancet 374, 1196–1208.

Gloor, G.B., Macklaim, J.M., Pawlowsky-Glahn, V., Egozcue, J.J., 2017. Microbiome datasets are compositional: and this is not optional. Front. Microbiol. 8, 2224.

Knight, R., Vrbanac, A., Taylor, B.C., Aksenov, A., Callewaert, C., Debelius, J., Gonzalez, A., Kosciolek, T., McCall, L.-I., McDonald, D., Melnik, A.V., Morton, J.T., Navas, J., Quinn, R.A., Sanders, J.G., Swafford, A.D., Thompson, L.R., Tripathi, A., Xu, Z.Z., Zaneveld, J.R., Zhu, Q., Caporaso, J.G., Dorrestein, P.C., 2018. Best practices for analyzing microbiomes. Nat. Rev. Microbiol. 16, 410–422.

Mandal, S., Van Treuren,W.,White, R.A., Eggesbø,M., Knight, R., Peddada, S.D., 2015. Analysis of composition ofmicrobiomes: a novel method for studyingmicrobial composition. Microb. Ecol. Health Dis. 26, 27663.

Morton, J.T., Sanders, J., Quinn, R.A., McDonald, D., Gonzalez, A., Vázquez-Baeza, Y., Navas- Molina, J.A., Song, S.J., Metcalf, J.L., Hyde, E.R., Lladser, M., Dorrestein, P.C., Knight, R., 2017. Balance trees reveal microbial niche differentiation. mSystems 2. https://doi. org/10.1128/mSystems.00162-16.

Morton, J.T., Marotz, C.,Washburne, A., Silverman, J., Zaramela, L.S., Edlund, A., Zengler, K., Knight, R., 2019. Establishing microbial composition measurement standards with reference frames. Nat. Commun. 10, 2719.

Weiss, S., Xu, Z.Z., Peddada, S., Amir, A., Bittinger, K., Gonzalez, A., Lozupone, C., Zaneveld, J.R., Vázquez-Baeza, Y., Birmingham, A., Hyde, E.R., Knight, R., 2017. Normalization and microbial differential abundance strategies depend upon data characteristics. Microbiome 5, 27.

Reviewer #2: Please correct some minor spelling mistakes while proofreading the final version of the manuscript - like "resent" instead of recent in the abstract.

7. PLOS authors have the option to publish the peer review history of their article (what does this mean?). If published, this will include your full peer review and any attached files.

Reviewer #1: No

Reviewer #2: No

---

## [Author Response · Author response to Decision Letter 1]

17 Nov 2021

Krzysztof Durkalec-Michalski, Ph.D

Academic Editor

PLOS ONE 

Yasuhiro Sasuga, Ph.D

Hachioji Center for Research & Development

B&S Corporation Co., Ltd.

17th November 2021

Dear Dr. Durkalec-Michalski

Response to reviewers’ comments [PONE-D-21-11683R1]

Thank you for reviewing our manuscript and for the helpful comments provided.

Please find enclosed our response to the comments raised. The line numbers provided here are based on our revised manuscript with tracked changes.

1. Additional Editor Comments

1. Line 27 – correct “resent” to “recent”.

 We appreciate the editor’s comment. We collected the text (line 21).

2. Line 66 – try to reduce even one "and" in this sentence - is repeated too often.

 We appreciate the editor’s comment. We collected the text (line 56-57).

3. Line 133 - The heading "TEST ARTICLE" is unclear. Consider Improving - Maybe "Supplementation". Likewise, in the later sections (e.g. lines 147-148) - the "test article" is misleading.

 We appreciate the editor’s comment. We collected the text from article to supplement (line 119, 120, and 138).

4. Lines 151-153 - Please add the necessary clarifications what exactly was and how "An official race" was conducted.

 We appreciate the editor’s comment. We inserted the text (line 142-147).

5. Line 164 – unclear part – authors wrote “and exercise intensity in order to evaluate exercise intensity” – do the authors mean “and exercise specificity in order to evaluate exercise intensity”? Please revise it.

 We thank the editor’s comment. We collected the text from “intensity” to “specificity” (line 158).

6. Lines 166-167 – unclear part – authors wrote “ exercise load of the race when participating in the race”. Maybe some of the terms "race" can be replaced with "exercise tests" or "competition".

 We appreciate the editor’s comment. We collected the text from “race” to “competition” (line 160).

7. The statements in the methodical part of biological material storage "as soon as possible" (lines 159 and 175) and are quite unstable and unclear - what does that mean? What was the criterion or the range of time?

 We appreciate the editor’s comment. We collected to “immediately” (line 168, line201). We also added the storage condition (line 168-169, Line 201-202).

8. Lines 216-241 - In the description of the statistical parts it should also be clearly indicated whether the minimum sample size has been achieved, bearing in mind some losses.

 This study was a pilot trial with a small number of participants. Since some data could not be obtained at some points, we described how to handle them (Line 211-213).

9. Line 354 - put "concentration" behind "arabinose".

 We have already described 17.00 ± 5.21 (SD) mmol / mol Cr as the arabinose concentration after ingestion of LEX (line 367).

10. Line 421 - the authors write "an official competition" - but what was the detailed specificity of these efforts. Were they real competitions in which competitors competed with other competitors (not participating in the project) or simulated competitions in which efforts imitating competitions were carried out?

It was the intention of real competitions. However, there is already research on athletic competition and gut microbiota, and this study was characterized by tracking changes in athletes' microbiota and urinary metabolites, so we changed the description to that (Line 441-443). 

11. Line 477 - try to use a more formal language and replace the word "fascinating".

 We appreciate the editor’s comment. We collected to “interesting” (line 490).

12. Lines 480-481 - the authors wrote "During long-distance running, the total energy of the body is very consumed" - this sentence is confusing. Do the authors mean that the body's energy expenditure is significant / elevated?

 We appreciate the editor’s comment. We collected to describe that long-distance runner need sustained energy (Line 493-495).

13. Line 484 - You should probably put "intake" behind “Fat and energy”

 We appreciate the editor’s comment. We put “intake” (Line 498).

14. Line 534 – change “is about 260km” to past tense.

 We appreciate the editor’s comment. We collected (line 546).

15. Lines 570-571 - the sentence, however, may be too optimistic despite the assurance that this is a hypothetical opinion of the authors. One should approach this even more carefully and point to "possible" or "potential" activation and that LEX ingestion "may be" mediated.

We appreciate the editor’s comment. We collected to “may be” (Line 580).

2. Reviewer 1

■ Most of the comments have been taken into account by the authors. However, I maintain my opinion that the article should include a list of the strains included in the product tested. As shown for example in the work of van Baarlen P et al. (van Baarlen P et al. Differential NF-kappaB pathways induction by Lactobacillus plantarum in the duodenum of healthy humans correlating with immune tolerance. Proc Natl Acad Sci U S A. 2009;106(7):2371-6.) killed strains have their specific physiological activities and it can be suspected that it is strain dependent. 

We appreciate the reviewer’s comment. We have described all strains (Line 123-125). Also, the description has been changed to novel genera to reflect the reclassification of the genus Lactobacillus (Zheng J et al. 2020).

（Reference）

・Zheng J, et al..: A taxonomic note on the genus Lactobacillus: Description of 23 novel genera, emended description of the genus Lactobacillus Beijerinck 1901, and union of Lactobacillaceae and Leuconostocaceae. Int J Syst Evol Microbiol. 2020 Apr; 70(4): 2782–2858.

■ Authors should clarify the answer to the compositionality problem. Microbiome data are compositional (relative) and give no information about absolute abundances, regardless of normalization procedures (Gloor et al., 2017). Therefore relevant statistical methods, e.g., based on log-ratios must be used, to avoid false-positive results (Knight et al., 2018; Mandal et al.,2015; Weiss et al., 2017). Information on the increase in abundance of a particular bacterial genus must be linked to the identification of the reference point (i.e. another genus) against which this occurred (Christensen et al., 2009;Morton et al., 2019, 2017). (Christensen et al., 2009;Morton et al., 2019, 2017).

We accepted this reviewer’s suggestion. We conducted linear discriminant analysis (LDA) Effect Size (LEfSe) method (Segata N et al. 2011) to detect the differences in bacterial taxa characterizing the groups. There was no change in the basic data interpretation due to the change in the analysis method.

Due to the change of analysis method, the description of the method (Statistical analysis section)　has been changed (Line 214-216).

Due to the change of analysis method, figures have also changed (Fig 2B, add Fig 2C, S5 Fig). Figure legends have also been changed (Line 292-296, Line 671-682).

The result of S2 Fig was partially inappropriate as a result of LEfSe analysis, so it was replaced with the F / B ratio (S2 Fig., and Line 648-655).

In the result section, the description has been changed based on the result of LEfSe analysis (Line 308-331, Line 422-436). 

The analysis of Parabacteroides distasonis, since LEfSe analysis was widely performed, exclusion by Smirnov-Grubbs test was not performed, and analysis was performed at n = 11.　 Therefore, the description of Smirnov-Grubbs test was deleted (Line 220, Line 432-433 Line 681-682).

Due to the change of analysis method, one reference has been added (reference No. 18) (Line 781-784). Therefore, subsequent reference numbers have changed.

（Reference）

Segata N, Izard J, Waldron L, Gevers D, Miropolsky L, Garrett WS, et al. Metagenomic biomarker discovery and explanation. Genome Biol. 2011;12(6):R60.

3. Additional changes

Organized the description in the Statistical analysis section.

◆Moved the software description to the end　（Line　240-242）.

◆Since ANOVA was a repeated measure, the description was changed from “one way ANOVA” to “repeated measures ANOVA” (Line 222, 302, 660).

◆The description of the F-test was missing, so we added it (Line 226-227).

Line 26: 16s → 16S

Line 195, 284: Unifrac → UniFrac

Line 405: tricarballyic → tricarballylic

Line 414: kynureic → kynurenic

Line 416: Added phylum and family of Parabacteroides distasonis.

Line 483: Actinovateria → Actinobacteria

Line 532: Additionaly → Additionally

Fig 1: Since the description of the number of analyzes in S5 FIg was missing, we added it. As a result, there are two “Fecal microbiota” in Data analysis, so we distinguished them (all and LEX).

We hope that we have addressed all the issues raised, and would be happy to clarify further on any other issues. Thank you for consideration of our manuscript for publication in your journal.

Yours sincerely,

Yasuhiro Sasuga

---

## [Decision Letter · Decision Letter 2]

6 Dec 2021

PONE-D-21-11683R2The impact of a competitive event and the efficacy of a lactic acid bacteria-fermented soymilk extract on the gut microbiota and urinary metabolites of endurance athletes: an open-label pilot studyPLOS ONE

 Dear Dr. Sasuga,

Thank you for resubmitting your manuscript to PLOS ONE and making your manuscript significantly improved. After careful consideration, we feel that it has merit but does not fully meet PLOS ONE’s publication criteria as it currently stands. The manuscript has been assessed by a statistical reviewer and his valuable comments significantly enrich the work. They should be carefully considered. Therefore, we invite you to submit a revised version of the manuscript that addresses the points raised during the review process.

We look forward to receiving your revised manuscript.

Kind regards,

Krzysztof Durkalec-Michalski, Ph.D

Academic Editor

PLOS ONE

Journal Requirements:

Reviewers' comments:

Reviewer's Responses to Questions

**Comments to the Author**

1. If the authors have adequately addressed your comments raised in a previous round of review and you feel that this manuscript is now acceptable for publication, you may indicate that here to bypass the “Comments to the Author” section, enter your conflict of interest statement in the “Confidential to Editor” section, and submit your "Accept" recommendation.

Reviewer #3: (No Response)

2. Is the manuscript technically sound, and do the data support the conclusions?

Reviewer #3: Yes

3. Has the statistical analysis been performed appropriately and rigorously? 

Reviewer #3: No

4. Have the authors made all data underlying the findings in their manuscript fully available?

Reviewer #3: Yes

5. Is the manuscript presented in an intelligible fashion and written in standard English?

Reviewer #3: No

6. Review Comments to the Author

Reviewer #3: Thoroughly proofread the manuscript. The results are not presented in an intelligent fashion. Pay attention to awkward sentences so the reader can comprehend the information being provided.

Minor revisions:

1- Line 218: Specify that the distribution was non-normal rather than “If failed.”

2- Line 219: Clarify that Scheffé refers to Scheffé’s multiple comparison method.

3- Line 221: Pearson’s correlation coefficients are used to illustrate the linear relationship between two variables. Additionally, graphs are provided. Pearson’s correlation coefficients are not obvious by inspecting graphs. Rephrase this sentence to improve clarity.

4- Lines 224-226: Clarify the use of the Wilcoxon signed-rank test. The term “non-parametric” can be dropped as a descriptor for Wilcoxon signed-rank and Freidman’s tests since no parametric versions of these tests exist.

5- Lines 227-230: Drop the portion of the sentence following the colon.

6- Line 240: Replace “background” with “baseline characteristics.”

7- Line 241-243: Upfront, state that the mean and standard deviation are being summarized. No need to repeat SD each time.

8- Line 245: Provide the percentage that corresponds to 8 of 13.

9- Lines 254-257: Clarify this sentence.

10- Line 266: Are the results statistically significant? If so, support the statement with a p-value.

11- The standard statistical term for average is mean.

12- Lines 274 -280: Clarify PRE_a, PRE_b, POST_a, POST_b.

13- Provide p-values to support statements indicating statistical significance.

14- Indicate the date range subjects were enrolled in the study.

15- The p-value associated with a correlation is a test of the null hypothesis: correlation equal to zero; however, the absolute magnitude of the coefficient indicates the strength of the linear relationship between two variables. In general, the strength or correlation coefficient is the more important statistic to focus on.

Below is a table for interpreting correlation coefficients:

Coefficient (absolute value) Interpretation

0.90 - 1.0 Very Strong

0.70 - 0.89 Strong

0.40 - 0.69 Moderate

0.10 - 0.39 Weak

less than 0.10 Negligible correlation

7. PLOS authors have the option to publish the peer review history of their article (what does this mean?). If published, this will include your full peer review and any attached files.

Reviewer #3: No

---

## [Author Response · Author response to Decision Letter 2]

10 Dec 2021

Krzysztof Durkalec-Michalski, Ph.D

Academic Editor

PLOS ONE 

Yasuhiro Sasuga, Ph.D

Hachioji Center for Research & Development

B&S Corporation Co., Ltd.

10th December 2021

Dear Dr. Durkalec-Michalski

Response to reviewers’ comments [PONE-D-21-11683R2]

Thank you for reviewing our manuscript and for the helpful comments provided.

Please find enclosed our response to the comments raised. The line numbers provided here are based on our revised manuscript with tracked changes.

1. Reviewer 3

■ Minor revisions:

■ 1- Line 218: Specify that the distribution was non-normal rather than “If failed.”

We appreciate the reviewer’s comment. We corrected to “non-normally distribution” (Line 218-219).

■ 2- Line 219: Clarify that Scheffé refers to Scheffé’s multiple comparison method.

We appreciate the reviewer’s comment. We corrected to “Scheffé’s multiple comparison method” (Lines 219, 630-631)

■ 3- Line 221: Pearson’s correlation coefficients are used to illustrate the linear relationship between two variables. Additionally, graphs are provided. Pearson’s correlation coefficients are not obvious by inspecting graphs. Rephrase this sentence to improve clarity.

We appreciate the reviewer’s comment. We rephrased this sentence (Lines 221-223).

■ 4- Lines 224-226: Clarify the use of the Wilcoxon signed-rank test. The term “non-parametric” can be dropped as a descriptor for Wilcoxon signed-rank and Freidman’s tests since no parametric versions of these tests exist.

We appreciate the reviewer’s comment. We dropped “non-parametric” (Lines 219, 228).

■ 5- Lines 227-230: Drop the portion of the sentence following the colon.

We appreciate the reviewer’s comment. We dropped the portion of the sentence following the colon (Lines 231-233).

■ 6- Line 240: Replace “background” with “baseline characteristics.”

We appreciate the reviewer’s comment. We replaced “background” with “baseline characteristics.” (Line 242).

■ 7- Line 241-243: Upfront, state that the mean and standard deviation are being summarized. No need to repeat SD each time.

We appreciate the reviewer’s comment. We corrected (Lines 243-245).

■ 8- Line 245: Provide the percentage that corresponds to 8 of 13.

We appreciate the reviewer’s comment. We described the percentage　（Line 248）.

■ 9- Lines 254-257: Clarify this sentence.

We appreciate the reviewer’s comment.　We rephrased this sentence to improve clarity (Lines 257-259).

■ 10- Line 266: Are the results statistically significant? If so, support the statement with a p-value.

We appreciate the reviewer’s comment.　We described p-value (Lines 271-273).

■ 11- The standard statistical term for average is mean.

We appreciate the reviewer’s comment. We replaced “average” with “mean.” (Lines 278, 298, 508, 525, 622, 629, 635, 652). 

■ 12- Lines 274 -280: Clarify PRE_a, PRE_b, POST_a, POST_b.

We appreciate the reviewer’s comment. It has described in materials & methods. However, we added explanations to improve clarity (Lines 280-281).

■ 13- Provide p-values to support statements indicating statistical significance.

We appreciate the reviewer’s comment.　We described p-value (Lines 310, 320, 352, 360-361, 363-366).

■ 14- Indicate the date range subjects were enrolled in the study.

We appreciate the reviewer’s comment. We described the date range subjects were enrolled (Line 135).

■ 15- The p-value associated with a correlation is a test of the null hypothesis: correlation equal to zero; however, the absolute magnitude of the coefficient indicates the strength of the linear relationship between two variables. In general, the strength or correlation coefficient is the more important statistic to focus on.

■ Below is a table for interpreting correlation coefficients:

■ Coefficient (absolute value) Interpretation

■ 0.90 - 1.0 Very Strong

■ 0.70 - 0.89 Strong

■ 0.40 - 0.69 Moderate

■ 0.10 - 0.39 Weak

■ less than 0.10 Negligible correlation 

We appreciate the reviewer’s comment. Focusing on the correlation coefficient, we described the coefficient interpretation (Lines 38, 376, 399, 405). 

We dropped “significant” (Lines 378, 383, 388).

We added coefficient (absolute value) (Lines 401-403, 406-409).

2. Additional changes

Organized the description in the Statistical analysis section.

Line 225: relationship → associations

Line 279, 284: delete “SD”

Line 348: delete “average”

Line 352: delete “SD”

Lines 407-408: delete “phenylalanine and tyrosine metabolites”.

Lines 408-409: delete “tryptophan metabolites”.

Line 409: delete “uracil”

Line 645: 9 → nine

Multi-panel Figures: place all panels from a multi-part figure into a single page and single file (Fig 1, Fig 2). 

We hope that we have addressed all the issues raised, and would be happy to clarify further on any other issues. Thank you for consideration of our manuscript for publication in your journal.

Yours sincerely,

Yasuhiro Sasuga

---

## [Decision Letter · Decision Letter 3]

10 Jan 2022

The impact of a competitive event and the efficacy of a lactic acid bacteria-fermented soymilk extract on the gut microbiota and urinary metabolites of endurance athletes: an open-label pilot study

PONE-D-21-11683R3

Dear Dr. Sasuga,

We’re pleased to inform you that your manuscript has been judged scientifically suitable for publication and will be formally accepted for publication once it meets all outstanding technical requirements.

Kind regards,

Krzysztof Durkalec-Michalski, Ph.D

Academic Editor

PLOS ONE

Additional Editor Comments (optional):

Reviewers' comments:

Reviewer's Responses to Questions

**Comments to the Author**

1. If the authors have adequately addressed your comments raised in a previous round of review and you feel that this manuscript is now acceptable for publication, you may indicate that here to bypass the “Comments to the Author” section, enter your conflict of interest statement in the “Confidential to Editor” section, and submit your "Accept" recommendation.

Reviewer #3: All comments have been addressed

2. Is the manuscript technically sound, and do the data support the conclusions?

Reviewer #3: (No Response)

3. Has the statistical analysis been performed appropriately and rigorously? 

Reviewer #3: (No Response)

4. Have the authors made all data underlying the findings in their manuscript fully available?

Reviewer #3: (No Response)

5. Is the manuscript presented in an intelligible fashion and written in standard English?

Reviewer #3: (No Response)

6. Review Comments to the Author

Reviewer #3: (No Response)

7. PLOS authors have the option to publish the peer review history of their article (what does this mean?). If published, this will include your full peer review and any attached files.

Reviewer #3: No

---

## [Editor Report · Acceptance letter]

12 Jan 2022

PONE-D-21-11683R3 

The impact of a competitive event and the efficacy of a lactic acid bacteria-fermented soymilk extract on the gut microbiota and urinary metabolites of endurance athletes: an open-label pilot study 

Dear Dr. Sasuga:

I'm pleased to inform you that your manuscript has been deemed suitable for publication in PLOS ONE. Congratulations! Your manuscript is now with our production department. 

Kind regards, 

on behalf of

Dr. Krzysztof Durkalec-Michalski 

Academic Editor

PLOS ONE